# Unpicking the Secrets of African Swine Fever Viral Replication Sites

**DOI:** 10.3390/v13010077

**Published:** 2021-01-08

**Authors:** Sophie-Marie Aicher, Paul Monaghan, Christopher L. Netherton, Philippa C. Hawes

**Affiliations:** 1African Swine Fever Vaccinology Group, The Pirbright Institute, Ash Road, Pirbright, Woking, Surrey GU24 0NF, UK; sophie-marie.aicher@pasteur.fr (S.-M.A.); christopher.netherton@pirbright.ac.uk (C.L.N.); 2Bioimaging, The Pirbright Institute, Ash Road, Pirbright, Woking, Surrey GU24 0NF, UK; paul.monaghan@csiro.au

**Keywords:** African swine fever virus, viral factories, viral proteins, click IT chemistry, STED microscopy, membrane assembly intermediates, electron tomography

## Abstract

African swine fever virus (ASFV) is a highly contagious pathogen which causes a lethal haemorrhagic fever in domestic pigs and wild boar. The large, double-stranded DNA virus replicates in perinuclear cytoplasmic replication sites known as viral factories. These factories are complex, multi-dimensional structures. Here we investigated the protein and membrane compartments of the factory using super-resolution and electron tomography. Click IT chemistry in combination with stimulated emission depletion (STED) microscopy revealed a reticular network of newly synthesized viral proteins, including the structural proteins p54 and p34, previously seen as a pleomorphic ribbon by confocal microscopy. Electron microscopy and tomography confirmed that this network is an accumulation of membrane assembly intermediates which take several forms. At early time points in the factory formation, these intermediates present as small, individual membrane fragments which appear to grow and link together, in a continuous progression towards new, icosahedral virions. It remains unknown how these membranes form and how they traffic to the factory during virus morphogenesis.

## 1. Introduction

African swine fever (ASF) is a lethal haemorrhagic disease of domestic pigs and wild boar, with a mortality rate of up to 100%. Since 2007, the disease has spread across Eurasia and led to huge losses of wild and domestic animals, with a concomitant devastating effect on the pork industry. The causative agent of the disease is African swine fever virus (ASFV), the only member of the Asfarviridae family and a member of the nucleocytoplasmic large DNA viruses (NCLDV) group, which also includes vaccinia virus, mimivirus, faustovirus and pacmanvirus [1,2]. Whereas ASFV shares its genomic organization with poxviruses, the icosahedral symmetry of the virion resembles Iridoviridae, which suggests a common ancestor for these viruses [3,4,5].

ASF virions are multi-layered structures approximately 200 nm in diameter, comprising a DNA-containing nucleoid, which is surrounded by the core shell or matrix. This is in turn encompassed by an inner membrane, with the capsid built around this internal envelope. A further envelope is acquired as the virus buds at the host cell plasma membrane, but virus with and without this external envelope are infectious [6]. Mass spectrometry analysis of highly purified extracellular virions revealed 68 viral proteins including structural proteins, enzymes and factors necessary for viral transcription, DNA repair and protein modification, as well as some viral proteins involved in host defence evasion. Host proteins were also identified, including a number related to actin motility, and it is thought that these host proteins are most likely recruited during virus budding at the host cell plasma membrane [7].

In host cells, the major site of genome synthesis and virion assembly is the virus factory. The factories made by ASFV, and other members of the NCLDV, show high levels of organization and complexity. Viral factories generated in protozoa and marine algae are similar to those formed in higher vertebrates, suggesting a need for factories evolved early during NCLDV evolution [8,9]. It is possible that NCLDV infection stimulates innate cellular responses, as described in other virus infections [10,11,12] that can then be hijacked to generate sites for replication and assembly. ASFV encodes its own transcriptional machinery that is packaged into the virion allowing transcription immediately after virus entry. Gene expression occurs in four stages: immediate-early and early before the onset of viral DNA replication, followed by intermediate and late which are dependent on early viral protein synthesis [13]. Virus morphogenesis starts in the perinuclear factory with the accumulation of membrane assembly intermediates thought to be derived from the endoplasmic reticulum (ER) [14,15]. During deposition of the capsid proteins and core shell proteins on the outer and inner faces of these membranes, the assembly intermediates form their characteristic icosahedral shape. After proteolytic cleavage of the core polyproteins, the last step is the incorporation and condensation of the nucleoid [16] before virions leave the factory and egress to the plasma membrane [17].

The morphology of virus factories changes during the course of infection and the mechanisms that govern the dramatic rearrangement of the host cell and the association with host membranes remain unknown [18]. In addition to genome replication, virion assembly and morphogenesis, viral factories are likely to be the sites of late viral transcription and translation. For poxviruses, it has been identified that viral factories are greatly compartmentalized to facilitate intermediate and late viral gene expression and subsequent protein synthesis in a highly coordinated, spatially separated fashion [19]. The ASFV factory is less well characterized. However, compartmentalization of late viral gene transcription and host translation initiation factors has been reported [20].

The structure of extracellular ASFV has recently been described by cryo-electron microscopy [21,22,23]. The capsid structure has been solved to 4.1 Angstroms and consists of one major (p72) and four minor capsid proteins (M1249L, p17, p49 and H240R). The minor capsid proteins have been shown to sit just below the outer capsid shell, and it is suggested that they form a framework within which the capsomers anchor into the inner membrane via the membrane protein p17 [23].

Detailed identification of the ASFV protein capsid structure opens up new possibilities for vaccine development. However, it does not address the question of how the membrane assembly intermediates form and how they travel into the factory during morphogenesis. Evidence for the ER being the origin of these intermediates is based on ER markers close to viral membranes [15,24]. It has been reported that some NCLDVs assemble via the membrane rupture model [25,26,27], which has recently superseded the previously held belief that enveloped viruses acquire their membrane by either budding from an existing cellular membrane to produce a single viral membrane or wrapping by collapsed cell membrane cisternae to produce a double-membraned virion. This new process of viral membrane formation has been described for vaccinia virus, mimivirus and latterly for ASFV. Biochemical evidence for these ASFV membranes being derived from ER has been presented. However, continuities between the ER and membranes in the factory were not seen [14]. A model describing ruptured ER forming open membrane curls which become targeted to the factory and coated with capsid is proposed although there is much detail yet to be defined.

In order to detail the pathway by which the viral membrane assembly intermediates form, and how they accumulate in the factory, it is important to catch the early stages of this process. In this study, we use confocal and super-resolution microscopy alongside electron microscopy and tomography to analyze the viral factory with respect to the protein and membrane compartmentalization and organization. We use high-pressure freezing to show that early factories contain individual membrane fragments which grow into curved membranes and link together before capsid and core protein deposition and further virion maturation.

## 2. Materials and Methods

### 2.1. Cell Culture, Virus and Antibodies

African green monkey kidney fibroblast (Vero) cells from the European Collection of Authenticated Cell Culture (Cat. 84113001) were cultured in Dulbecco’s Modified Eagle’s Medium (DMEM) with GlutaMax-I (Gibco/Thermo Fisher Scientific, Waltham, MA, USA) supplemented with 10% fetal bovine serum (FBS), 100 units/mL penicillin and 100 μg/mL streptomycin at 37 °C and 5% CO_2_. The Vero-adapted Badajoz 1971 (Ba71v) strain of African swine fever virus (ASFV) has been described previously [28] and infections were performed at a multiplicity of infection (MOI) of 3. Antibodies against ASFV have been described previously: mouse monoclonal 17LD3 anti-p72 [29], mouse monoclonal 4H3 anti-p72 [30], rabbit RB7 anti-p54 [31], rabbit TW34 anti-p34 [3,31], anti-ASFV serum MI92 (The Pirbright Institute), rabbit SB11 anti-pE120R [17], mouse polyclonal Y1 anti-pP1192R [32], Alexa488 goat anti mouse, Alexa488 goat anti rabbit, Alexa555 goat anti mouse (Life Technologies/Thermo Fisher Scientific, Waltham, MA, USA), Alexa555 Azide triethylammonium salt (Life Technologies/Thermo Fisher Scientific, Waltham, MA, USA), goat anti-mouse 10 nm gold, and goat anti-rabbit 5 nm gold (BBI Solutions, Crumlin, UK).

### 2.2. Immunofluorescence and Confocal/STED Microscopy

Cells were grown on glass coverslips and subsequently fixed in 4% paraformaldehyde for 30 min. After a 15 min permeabilization with 0.2% Triton X-100, cells were blocked with 0.2% gelatin from cold water fish skin, 0.2% natrium azide and 10% goat serum in 1x TBS for 1 h, incubated with primary antibody diluted in blocking buffer for 30 min then secondary antibody for 30 min in the dark. Cells were washed with PBS-A after every step. All above solutions were sourced from Sigma/Merck, Darmstadt, Germany. Cellular and viral DNA were stained with DAPI (for confocal imaging) or ToPro-3 (for STED imaging) (Life Technologies/Thermo Fisher Scientific, Waltham, MA, USA) before the cells were washed with ultrapure water and mounted in ProLong Diamond antifade mountant (Life Technologies/Thermo Fisher Scientific, Waltham, MA, USA). It was not possible to use DAPI for STED imaging as it is excited by the 592 nm depletion laser creating excessive background when collecting data. Both DNA stains were found to be equally effective at identifying nuclei and viral factories.

Images were acquired with or without activated STED function on a Leica TCS SP8 STED 3X microscope equipped with white light excitation laser, 592 and 660 nm depletion lasers. Confocal images were taken at 1024 × 1024 pixel format with 4× frame averaging, STED images were adjusted to 20 × 20 μm or 10 × 10 μm pixel size and acquired with 4× frame averaging and 3× line accumulation. Overlay pictures of single-channel images were digitally created in Adobe Photoshop CS6 13.0 (San Jose, CA, USA). STED images were subject to deconvolution in Huygens Professional software 18.04 (Scientific Volume Imaging, Netherlands) following the Deconvolution Wizard with a theoretical PSF.

### 2.3. Analysis of Protein and DNA Localization Using Click IT Chemistry

For analysis of nascent proteins, medium was removed from plated, infected cells and replaced with L-methionine-free DMEM (Gibco/Thermo Fisher Scientific, Waltham, MA, USA) supplemented with 2% dialyzed FBS and 100 units/mL penicillin and 100 μg/mL streptomycin at indicated time points. Following 45 min of starvation at 37 °C and 5% CO_2_, L-homopropargylglycine (HPG, Life Technologies/Thermo Fisher Scientific, Waltham, MA, USA) was added at a final concentration of 0.5 mM for 30 min (for microscopy) or 1 mM for 1 h (for in-gel fluorescence and western blot, Appendix A) in L-methionine-free DMEM. As the control, standard DMEM was added instead of HPG. After staining for ASFV antigens, HPG pulse-labelled cells were subject to Click IT reaction in a reaction buffer containing Alexa Fluor 555-azide as previously described [33]. The reaction was allowed to proceed for 2 h at RT in the dark before coverslips were stained with ToPro-3 and mounted in Prolong Diamond. For analysis of DNA synthesis, plated cells were cultured in the presence of 5-ethynyl-2′-deoxyuridine (EdU, Sigma/Merck, Darmstadt, Germany) at a final concentration of 5 µM in standard culture media for two hours prior to fixation. EdU-labelled cells were then quenched with 100 mM glycine for 5 min before permeabilization, labelling and 2 h in the same Click IT reaction buffer containing Alexa Fluor 555-azide as described previously [33].

### 2.4. Preparation of Thawed Cryo-Sections for TEM

Infected cell monolayers were fixed at room temperature in immunogold fixative (2% paraformaldehyde with 0.05% glutaraldehyde in a phosphate buffer), gently scraped from the substrate, spun into a pellet and re-suspended in 12% gelatin in PBS. The cell pellet was cooled and cut into 1 mm^3^ blocks which were infiltrated with 2.3 M sucrose at +4 °C overnight. Blocks were attached to cryo-pins (Leica Microsystems, Milton Keynes, UK) using 2.3 M sucrose and frozen in liquid nitrogen. The 60 nm sections were cut at −120 °C (Leica Microsystems UCT cryo-ultramicrotome, Milton Keynes, UK) and allowed to reach room temperature before labelling.

### 2.5. Immunogold Labelling of Thawed Cryo-Sections

Sections on TEM grids were incubated in blocking buffer (1% bovine serum albumin, 0.5% FBS, 1% fish gelatin in Dulbecco’s PBS, Sigma/Merck, Darmstadt, Germany) for 30 min before being transferred to 50 µL droplets of primary antibody in a moist chamber and left overnight at room temperature. After PBS washes, sections were incubated in species specific gold conjugate for 90 min in a moist chamber. Samples were washed in 2% glutaraldehyde to ensure the colloidal gold was fixed in place, and stained with 3% aqueous uranyl acetate (Agar Scientific, Stansted, UK) before being coated in a thin layer of methyl cellulose to provide support whilst imaging.

### 2.6. High-Pressure Freezing and Freeze Substitution of Cell Monolayers

Infected cells on 3 mm sapphire glass coverslips were frozen in a Bal-tec HPM010 HPF according to [34] before being transferred to a pre-cooled (−160 °C) automatic freeze substitution unit (AFS, Leica Microsystems, Milton Keynes, UK) and allowed to warm to −90 °C over a period of 3.5 h. FS protocol has been described elsewhere [34]. Briefly, cold (−90 °C) freeze substitution medium containing 0.2% uranyl acetate in acetone was carefully added to the samples and left for 60 min before warming to −50 °C. Samples were taken through a graded series of resin concentrations (Lowicryl HM20, Agar Scientific, Stansted, UK, in acetone), placed in Leica flat embedding moulds (cells facing upwards) and polymerized under UV light at −50 °C for 48 h and room temperature for a further 48 h. Thin sections (60 nm) of the resulting resin blocks were cut (Leica Microsystems, UC6 ultramicrotome, Milton Keynes, UK) and imaged at 100 kV in a FEI Tecnai12 transmission electron microscope (Thermo Fisher Scientific, Waltham, MA, USA) with Tietz F216 2K × 2K CCD camera (TVIPS, Munich, Germany) after counter staining with uranyl acetate and lead citrate (EM Stain, Leica Microsystems, Milton Keynes, UK).

### 2.7. Chemical Fixation of Cell Monolayers for TEM

Infected Vero cells on 13 mm Thermanox coverslips (Nunc, from Thermo Fisher Scientific, Waltham, MA, USA) were prepared for TEM using a standard protocol, detailed in [35]. Briefly, after 60 min in 2% glutaraldehyde in phosphate buffer and 60 min in 2% aqueous osmium tetroxide, coverslips were taken through a graded series of ethanols to 100% over a period of 60 min. Samples were infiltrated with epoxy resin and polymerized at 60 °C overnight. Sixty nanometer sections were cut, further stained with uranyl acetate and lead citrate and imaged at 100 kV in a FEI Tecnai 12 TEM using a Tietz F214 CCD camera. All solutions and EM consumables were sourced from Agar Scientific, Stansted, UK.

### 2.8. Electron Tomography (ET) of Infected Cell Monolayers

Thick sections (250–300 nm) of infected, high-pressure frozen/freeze substituted Vero cells were cut and collected on Formvar coated 100 mesh hexagonal thin bar gold TEM grids (Agar Scientific, Stansted, UK). After the addition of 10 nm colloidal gold fiducial markers (BBI Solutions, Crumlin, UK), unstained samples were imaged at 120 kV in a FEI Tecnai 12 using a dual axis tomography holder (Model 2040, Fischione Instruments Inc., Export, PA, USA). After a suitable area was found, a tilt series was collected recording an image every 1° over a range of ± 60°. The grid was rotated through 90° and another tilt series was collected creating a 121 image dataset. The FEI data collection module of the Inspect3D software package was used to automatically collect each series. Images within each series were aligned and used to create a tomogram (3D distribution of stain density). The two tomograms were then combined to produce a single, dual axis reconstruction of the region of interest. Image alignment, tomogram production and tomogram combination were carried out using the IMOD software package [36]. The 3D reconstructions were segmented by modeling membrane assembly intermediates using IMOD. Membranes were traced by hand on consecutive Z slices within a tomogram using a Wacom Cintiq 21 UX (Saitama, Japan) interactive graphics tablet and pen.

## 3. Results

ASF virus factories appear as perinuclear structures surrounded by host cell organelles when imaged using differential interference contrast (DIC) (Figure 1a, top) and their extent are clearly defined by the presence of extra nuclear DNA (Figure 1a, bottom, arrow). Confocal microscopy analysis of the localization of the major capsid protein p72 (*B646L* gene) and the internal envelope protein p54 (*E183L* gene) within the virus factory revealed two different patterns. p54 signal resolved as a pleomorphic structure within the virus factory with p72 signal resolving as puncta that likely represent individual newly forming or formed virions that were closely associated with the p54 signal (Figure 1b).

However, at 200 nm in diameter, the virions are at the limit of resolution of a confocal microscope. STED microscopy increased the achievable resolution from these immunofluorescence labelled samples so that it was possible to clearly resolve the p72 capsid ring surrounding the central core of each virion particle (Figure 1c,d,g). Although there was a clear association between p72 and p54 labelling the signals did not co-localize. The increased resolution offered by STED revealed an intricate reticular pattern within the p54 labelling, previously seen as a continuous ribbon by confocal microscopy (Figure 1b). This pattern revealed a complex network of p54 positive structures (Figure 1e,h), clearly associating with new virions (Figure 1f,i). The spatial relationship between p72 and p54 was similar to that seen after labelling with a polyclonal sera which recognises core shell protein p34 and the polyprotein pp220 (*CP2475L* gene) from which it is derived (Appendix A).

pE120R binds to the major capsid protein p72 and has an essential role in the release of the mature virions from the factory and their transport to the plasma membrane [6]. pE120R should, therefore, co-localize with p72 and represent another marker for the capsid of ASFV virions. STED analysis in Figure 2b revealed similar doughnut-shaped structures to those previously seen for p72 (Figure 2a) and the single capsid structures co-localized with the p72 virions (Figure 2c). Intriguingly, the fluorescence signals did not completely overlap, and green-red double-stained virions became visible rather than uniformly yellow ones. This might suggest that both proteins are assembled into the capsid in adjacent yet distinct sites (Figure 2c insets). However, the possibility that this was an artefact of labelling with two capsid antibodies simultaneously cannot be ignored.

Although viral DNA is generally used to define the location of viral factories, viral structural proteins localize to defined areas within that virus factory. The *P1192R* gene of ASFV encodes for a topoisomerase II, an enzyme that modulates the topological state of DNA during replication and/or transcription. Labelling with an anti-pP1192R antiserum confirmed that the protein localized to virus factories [32]. However, STED imaging revealed two distinct signals within factories (Figure 2d). The brightest signal co-distributed to the mass of assembling virions, visualized using anti-pE120R antibody (Figure 2e). However, a weaker signal existed throughout the remainder of the virus factory, as delineated using ToPro3 labelling.

Super-resolution microscopy identified sub-divisions of the ASF virus factory and in particular the association between assembled virions and an apparent accumulation of viral proteins such as p54 (Figure 1) and p34 (Appendix A). The location of viral RNA around the replication site [20] suggests that virus factories may contain specialized domains for synthesizing viral proteins. Click-IT chemistry was, therefore, used to gain an understanding of the effect of ASFV on protein synthesis. Nascent proteins can be visualized in cells by incorporating the alkyne containing methionine analogue HPG, followed by a covalent reaction with a probe tagged with azide (Appendix A). Newly synthesized proteins became visible in all cells and followed the previously reported pattern [33] of cytoplasmic staining with accumulation at other cellular organelles—in our case, mitochondria, and abundant nuclear staining concentrating in the nucleoli (Figure 3a).

Interestingly, in Vero cells infected for 16 hpi, the cytoplasm and nuclear labelling appeared diminished compared to neighboring uninfected cells, and a perinuclear concentration of nascent proteins co-incident with the viral factory was seen (Figure 3b,c) suggesting incorporation into nascent viral proteins.

The accumulation of newly synthesized proteins in the factory resembled the ribbon-like protein structure previously seen for p54 (Figure 1c) and p34 (Appendix A). Therefore, we applied STED microscopy to HPG-labelled cells at 16 hpi and probed in parallel for p72, p54 and p34 (Figure 3d–i, Appendix A). Super-resolution analysis revealed that the nascent protein accumulation in the viral factory co-localized with both p54 and p34 (Figure 3g–i, Appendix A) and mature p72 virions again wrapped around the ribbon of reticular structures (Figure 3d–f). These findings indicate that late viral proteins are synthesized in the factory center or shuttled to it immediately after synthesis, supporting our hypothesis that ASF virions are formed inside the protein ribbon and presented on its edges when mature and ready to leave the factory.

The intricate reticular pattern of p54 (Figure 1e,h), p34 (Appendix A) and HPG labelling (Figure 3d,g) in STED images is strikingly similar to the arrangement of membrane assembly intermediates, previously only described from electron microscope images [14]. To confirm this, we conducted immunogold experiments, labelling thawed cryo-sections with antibodies against p72 and p54.

The p72 antibody clearly localized to whole virions (Figure 4a) and p54 antibody was precisely restricted to the region immediately surrounding the virions containing the membrane assembly intermediates (Figure 4b). The similarity in the relationship between p72 and p54 signal when observed by indirect immunofluorescence with STED microscopy or immunogold labelling in the electron microscope was striking. To our knowledge, this is the first time the compartment within the factory associated with p54 has been identified in detail using light microscopy.

We then used high-pressure freezing followed by freeze substitution to prepare infected cell monolayers to give a more accurate representation of sub-cellular detail than standard chemical fixation [34]. Images of thin sections of this material clearly showed whole virions in either mature or immature stages surrounding membrane assembly intermediates amidst an electron dense protein/DNA accumulation (Figure 4c, black arrows) in the center of the factory. These membrane assembly intermediates took many forms, from partly formed icosahedral structures (Figure 4d, yellow arrow) with the capsid protein layer accumulating on the outer face and inner core shell proteins assembling on the inner face, to individual curved membranes (Figure 4d, blue arrows) and, interestingly, membrane fragments (Figure 4d, pink arrows) both apparently lacking the dense capsid or core shell protein layers. These curved membranes and membrane fragments vary in size and appear to have a clear beginning and end, suggesting that they have open ends free in the cytoplasm of the cell. Lipid bilayers in an aqueous solution close if ruptured, due to the hydrophobic nature of the fatty acids inside the bilayer. It is possible that these membranes are connected to each other and/or host cell compartment membranes and the appearance of free ends is a consequence of viewing a thin 2D section through the factory. To investigate this possibility, thicker sections of this same material were imaged in 3D using electron tomography.

All forms of membrane assembly intermediates, plus whole mature and immature virions, are visible in tomographic reconstructions of thick sections through HPF prepared infected Vero cells. Figure 5a,b are virtual slices picked out from a representative tomogram (Appendix A). Membrane fragments are visible throughout the reconstructed volume (Figure 5a,b, pink arrows) as are larger, curved membrane assembly intermediates (Figure 5a, black arrows). These membrane fragments and curved membranes do not appear to have the thick capsid protein layer on their outer surface yet. Any membrane assembly intermediates with characteristic vertices (the beginnings of the icosahedral shape) have capsid protein on their outer surface, and core shall proteins on their inner surface (Figure 5b, blue arrows). Formation of new virions is, therefore, likely to follow an ordered pathway starting with small membrane fragments, which grow or coalesce into curved membranes, which in turn become icosahedral following the deposition of capsid and core protein on the outer and inner faces of the membranes respectively.

The 3D model of part of this reconstruction (Figure 5c,d) shows a circular, part-formed virion connected by a membrane ribbon to another part-formed virion (Figure 5c,d, white arrow). The icosahedral shape had not started to form yet. Isolated membrane fragments are also seen, some small (Figure 5c, pink arrows) and some slightly larger (Figure 5c,d, light blue arrows), suggesting that these fragments grow or coalesce in the factory. When the model is viewed from a different angle, these fragments appear to be associated with each other, forming narrow stacks or ‘bridges’ linking curved membrane assembly intermediates (Figure 5d, dark blue arrows).

This pattern was seen again in a 3D model created from another tomogram (Figure 6a,b, Appendix A). In one highlighted area, membrane fragments formed stacks between a larger, curved section of membrane in what could be the start of virion formation (Figure 6c). When viewed from a different angle, these membranes appear to form a skeleton virus which could be the template for a curved pre-virion (Figure 6d).

Even following a synchronous infection protocol, it is common that cells within a population will become infected at different time points. For example, if cells are left for 18 h after initial infection and before fixation, a proportion of those cells will be infected for less than 18 h, possibly due to delays in virus attachment, entry or uncoating. The factory depicted in Figure 7 was part of a cell from a monolayer infected for 18 h, yet it is clearly not as advanced as the factories in Figure 5 and Figure 6 in terms of the presence of curved membrane assembly intermediates and whole virions. Tomographic slices show the factory was in the earlier stages of formation (probably equivalent to 12–14 hpi) and contained membrane fragments and little else visible by EM (Figure 7a,b, Appendix A). When the model of part of the factory was viewed in 3D (Figure 7c,d) it was clear the fragments were discrete from one other, and the larger fragments were only slightly curved. These are most likely to be early stages in the formation of the curved membrane assembly intermediates seen in later factories; the discrete fragments develop into curved membrane assembly intermediates, which then go on to form icosahedral immature virions.

The segregation between factory and cytoplasm is known to occur early in infection from previous studies [37]. We, therefore, wanted to investigate earlier, less mature factories using confocal and electron microscopy to capture both protein/DNA localization and ultrastructural data. To identify early viral DNA synthesis, nucleotide analogues were incorporated into newly synthesized DNA strands which were subsequently covalently linked to a detection reagent using Click-IT chemistry. The nucleotide analogue EdU became incorporated into viral DNA in early factories (Figure 8a). At 10 hpi ASFV protein localized specifically to the factory when labelled with an anti-ASFV serum, although the exact identity of the protein remains unknown (Figure 8a,b). When samples containing EdU were labelled with anti-p54 antibody, known to be an intermediate rather than late protein, no p54 was seen in the factory (Figure 8c). TEM imaging of early factories (up to 12 hpi) (Figure 8d) revealed that host cell components had been excluded from the replication area, possibly due to the presence of DNA and viral proteins, and few if any membrane precursors were visible inside the factory. A tomographic reconstruction of part of the edge of a slightly more mature factory from the same sample shows an accumulation of small vesicles at the periphery, along with two vesicles and a few membrane fragments at the center (Figure 8e,f,g, Appendix A). On inspection, none of the early factories imaged demonstrated any evidence of ER membrane cisternae linking the cell cytoplasm with the center of the factory. Factories from cells infected for shorter time periods still need to be investigated, and could be key to understanding where the membrane fragments come from and how the membrane assembly intermediates form.

## 4. Discussion

Assembly of the intracellular viral membrane of the NCLDVs is still poorly understood. However, it is clear that in several viruses, it does not rely on the usual budding or wrapping events that occur during the assembly pathways of other viruses.

Recent publications have provided evidence for the incorporation of open, ruptured ER membrane or membrane vesicles derived from ER, into membrane sheets which then curl to form spherical viruses (in the case of vaccinia virus and mollivirus) or icosahedral viruses in the case of mimivirus [38,39] and this has also been suggested in the assembly of ASFV [14]. A process whereby ER membranes pinch off from cytoplasmic cisternae, rupture to form membrane curls which are then coated with capsid and viral core proteins and mature into icosahedral virions has been hypothesized [14].

Here we used super-resolution microscopy in combination with biorthogonal labelling, along with electron tomography, to characterize the development of ASFV factories and membrane assembly intermediates. Previous work suggested the recruitment of viral mRNA to the periphery of ASFV factories and the recruitment of eukaryotic initiation factors to the factory itself [20]. Here we show that the direct consequence of this recruitment is perhaps unsurprisingly the accumulation of newly synthesized protein within the virus factory. However, at least 90 viral proteins are expressed at late time points [40,41], many of which are not thought to be involved in viral replication or assembly and are not incorporated into assembling virions [7]. Therefore, it was striking that such an intense signal associated precisely with the membrane assembly intermediates as defined by p54 labelling. This suggests that the membrane assembly intermediates represent the core protein synthesis and assembly sub-structure within the virus factory. The function of the remainder of the factory as defined by DNA and weak P1192R labelling is unclear as FISH suggests that viral mRNA accumulates outside of the factory [20] and, therefore, these areas are unlikely to be dedicated to transcription. Biorthogonal labelling with ethynyl-nucleosides identified early replication sites consistent with small factories identified by EM that had few membrane assembly intermediates. These sites were positive for unknown viral protein(s) after labelling with an antisera from a recovered pig, but negative for late proteins such as p54, p72 and pp220. Antibodies to a number of different proteins have been identified in serum from recovered pigs [42] and further characterization of such anti-sera may identify novel viral proteins involved in factory development.

We chose to further probe the early stages of virion morphogenesis. However, before we investigated this, we took a step back to characterize the early membrane assembly intermediates in greater detail. Our data show what could be interpreted as a continuous progression of virus membrane assembly in the factory, starting with individual, un-linked membrane fragments, which grow to form curved membranes and link together creating ‘skeleton viruses’, which develop into spherical pre-virions before capsid and core proteins assemble on the outer and inner surface of the membranes to create icosahedral immature virions.

Naturally questions then arise regarding the origin of the assembly intermediates and how they travel into the factory. In order to investigate this, we had to step back again and image earlier factories. The early stages of factory formation can be identified in the fluorescence microscope by accumulation of nascent DNA. In the electron microscope, these areas appear empty of host cell components. Most membrane and cytoskeletal elements are absent, so if the membrane fragments are derived from the ER, they would have to traffic from the cell cytoplasm into the factory without using microtubules. At the periphery of one of these early factories, our data show an accumulation of small vesicles which could be derived from the ER. The ER is a large, dynamic structure within cells and fulfills many, diverse functions [43]. It is the site of most lipid synthesis and it generates new organelles, for example peroxisomes, by budding from specific domains along its length [44]. It is possible that the vesicles seen in our results could have been produced by a similar method in response to increased phospholipid synthesis due to infection. In the center of the factory, a small number of vesicles and individual fragments of membrane were seen. However, in this study, we did not observe any membrane vesicles in the space between the periphery and the center of the factory, which could have provided evidence for the membrane vesicles being the origin of virus membrane, as suggested for mimivirus [27]. Further, membrane fragments in the factory were small and did not have the acute curvature which would be expected of ruptured vesicles, as seen in the assembly of other NCLDVs. It seems unlikely, therefore, that the membrane fragments are derived from the small vesicles found at the periphery of the factory, without them going through major remodeling or disassembly into component parts whilst trafficking to the factory center.

When the viral protein p54 is repressed during infection, aberrant zipper-like membranes form from disrupted ER and collect around the outside of the factory, implying that p54, or at least intact ER, is crucial for membrane presence in the factory [45]. Although all cellular membranes are derived from existing membranes and not created from individual phospholipids and proteins [46], it is possible that the fragments which go on to form virions are created by de novo synthesis of membrane in the factory. Potentially, the factory could be thought of as a non-membrane bound compartment [47] containing the components needed for membrane synthesis and virus assembly. If this liquid phase is separate from the surrounding cytoplasm, both having different viscosities, mixing of components inside the factory could continue without spreading into the cytoplasm. These liquid-like compartments have been shown to exchange component molecules rapidly with the cytoplasm [48]. It is conceivable, therefore, that components needed for membrane biogenesis (for example fatty acid synthase, FASN) could be delivered via ER derived vesicles to the factory edge and diffuse into the assembly site. Upregulation of FASN, which catalyzes the synthesis of palmitate necessary for the production of phospholipids, has been described in HCV infected cells although it does not localize to the sites of replication, remaining in the same cytoplasmic compartment as in uninfected cells [49]. In DENV infection, however, FASN does locate to the replication area, and replication is reduced when pharmacological or RNAi inhibition of FASN is initiated [10]. Further analysis of the lipid profile and biogenesis pathways in ASFV infected cells is required to investigate how these membrane assembly intermediates form and how they travel into the factory during morphogenesis.

## 5. Conclusions

Viral factories are complex regions with clear compartmentalization of protein and membrane components.Super-resolution microscopy reveals the sub-structure within virions and factory compartments with the advantage over challenging electron microscopy techniques of identifying the location of specific proteins.Click IT chemistry is a valuable technique for investigating the presence of newly synthesized protein.Different forms of membrane assembly intermediates are seen in early factories by HPF EM; membrane fragments, larger curved membranes and protein-coated membranes with the beginnings of the icosahedral profile.It is still not known where the membrane fragments come from or how they travel into the factory.

## Figures and Tables

**Figure 1 viruses-13-00077-f001:**
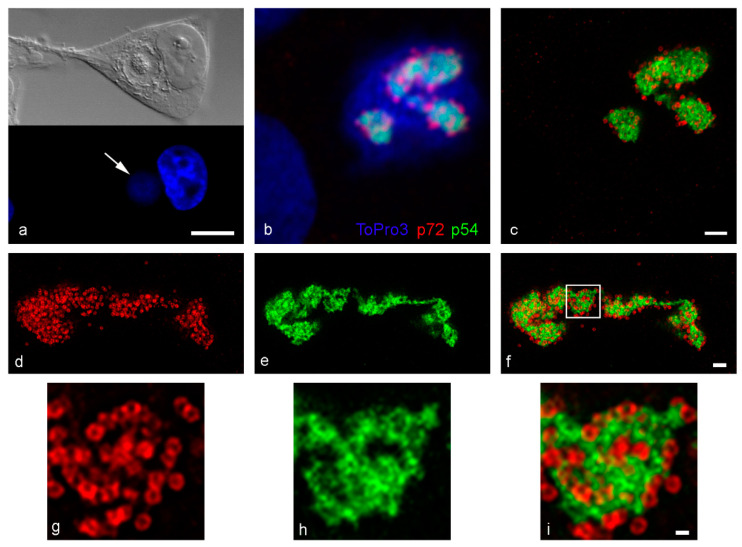
ASFV infected Vero cells by confocal and STED microscopy, fixed at 16–18 hpi. (**a**) DIC (top) imaging shows 18 hpi viral factories are distinct perinuclear structures containing viral DNA (DAPI, bottom, arrow). (**b**) p54 (intermediate protein, RB7 antibody) and p72 (late protein, 17LD3 antibody) localize to the viral factory at 16 hpi by confocal. (**c**) STED imaging of the same cell reveals more detail. (**d**,**g**) STED imaging of the major capsid protein p72 clearly reveals capsid rings. (**e**,**h**) STED imaging of p54 details an intricate reticular network not previously seen by confocal microscopy. (**f**,**i**) Overlay of p72 and p54 labelling confirms the close association between the proteins, but there is no co-localization. The capsid rings sit around the protein network. Scale bars: a = 10 µm, c, f = 1 µm, i = 200 nm.

**Figure 2 viruses-13-00077-f002:**
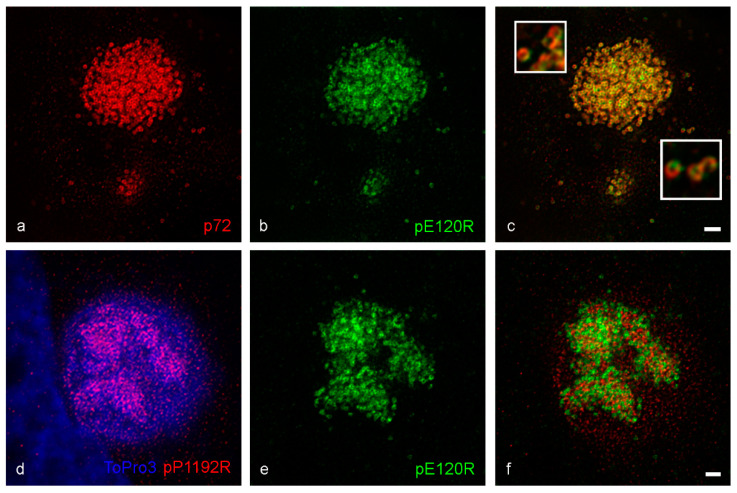
Localization of pE120R and pP1192 in ASFV infected Vero cells, fixed at 16 hpi. (**a**–**c**) STED imaging of antibodies against the major capsid protein p72 (17LD3) and binding partner pE120R (SB11). The overlay shows distinct regions of the virus capsid that are labelled with each antibody separately (**c**, insets). This is interesting, but it may be a labelling artefact as it does not correspond with the recently solved virus structure. (**d**–**f**) STED imaging of the topoisomerase antibody (Y1) alongside pE120R. Topoisomerase was present throughout the factory, as demarcated by ToPro3, but located mostly to the region containing pE120R (new virions). Scale bars: c, f = 1 µm.

**Figure 3 viruses-13-00077-f003:**
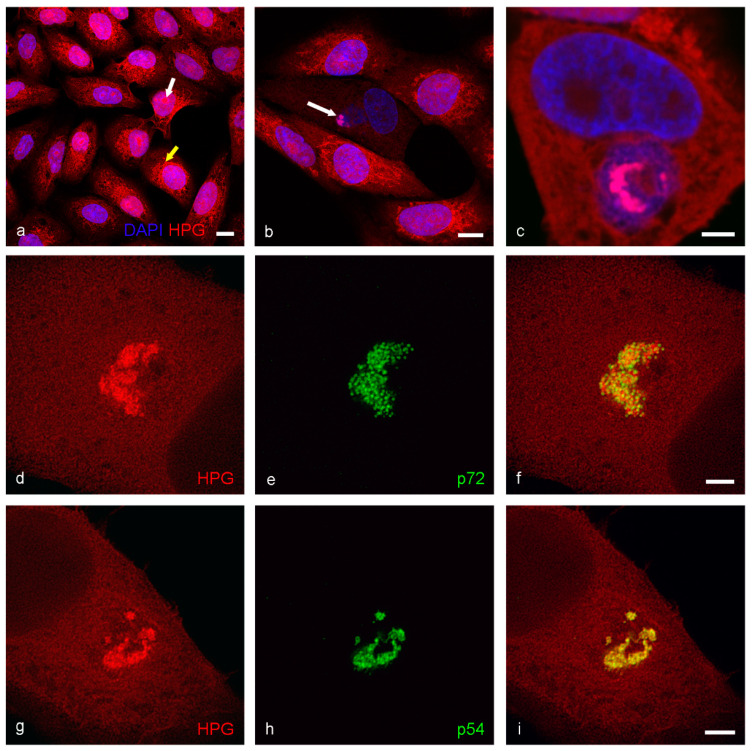
Incorporation of HPG into proteins in Vero cells. Cells were incubated with HPG for thirty minutes after forty-five minutes starvation in methionine free media, then fixed. Cells were subject to Click-IT reaction with azide conjugated to Alexa 555. Confocal image of uninfected (**a**) and 16 hpi (**b**,**c**) cells show an accumulation of nascent protein in cellular organelles (**a**, yellow arrow) and nucleoli (**a**, white arrow) and in the viral factory in infected cells (**b**, white arrow and **c**). By STED microscopy (**d**–**i**), the HPG labelling co-localized with both p72 (17LD3) and p54 (RB7) labelling. Scale bars: a, b = 10 µm, c, f, i = 3 µm.

**Figure 4 viruses-13-00077-f004:**
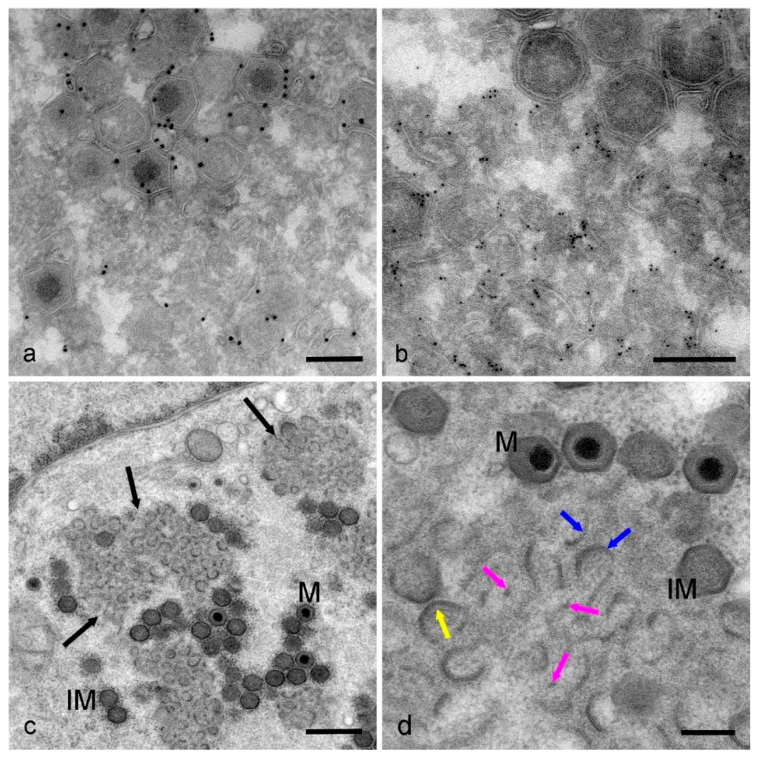
Transmission electron microscopy of viral factories in Vero cells at 18 hpi. Immunogold labelling of cryosections with antibodies against (**a**) p72 (4H3 antibody) with 10 nm gold and (**b**) p54 (RB7 antibody) with 5 nm gold. 10 nm gold clearly and distinctly localizes to whole virions whereas 5 nm gold localizes to the membrane assembly intermediates. (**c**) A low power image of high-pressure frozen (HPF) infected cells displays the different structures within a factory. Immature (IM) and mature (M) virions are visible surrounding darker areas of protein containing membrane assembly intermediates (black arrows). (**d**) At higher power, it is possible to identify different forms of membrane assembly intermediates; membrane fragments (pink arrows), larger curved membranes (blue arrows) and more mature intermediates with capsid and core protein and the beginnings of the icosahedral angles (yellow arrow). Scale bars: a, b, d = 200 nm, c = 500 nm.

**Figure 5 viruses-13-00077-f005:**
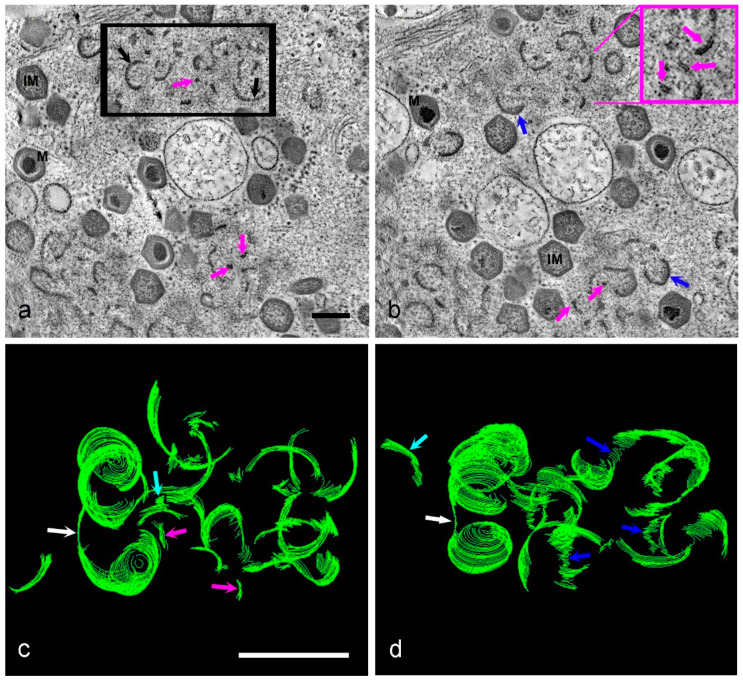
I. Tomographic reconstruction of HPF Vero cells at 18 hpi. (**a**,**b**) Two slices from the tomogram in Appendix A. High-pressure freezing reveals the existence of small membrane fragments (pink arrows) and the clarity afforded by tomography shows the capsid and core proteins accumulating on the outer and inner faces of the later membrane assembly intermediates (blue arrows). The area within the black box in (**a**) was modeled in IMOD and then viewed from two different angles (**c**,**d**). Small, apparently individual, membrane fragments can be seen (pink arrows) alongside slightly larger curved membranes (light blue arrows) with a connection visible between curved pre-virions (white arrows). When viewed from a different angle, the membrane fragments appear to form links or bridges between larger intermediates (dark blue arrows). Scale bars = 200 nm.

**Figure 6 viruses-13-00077-f006:**
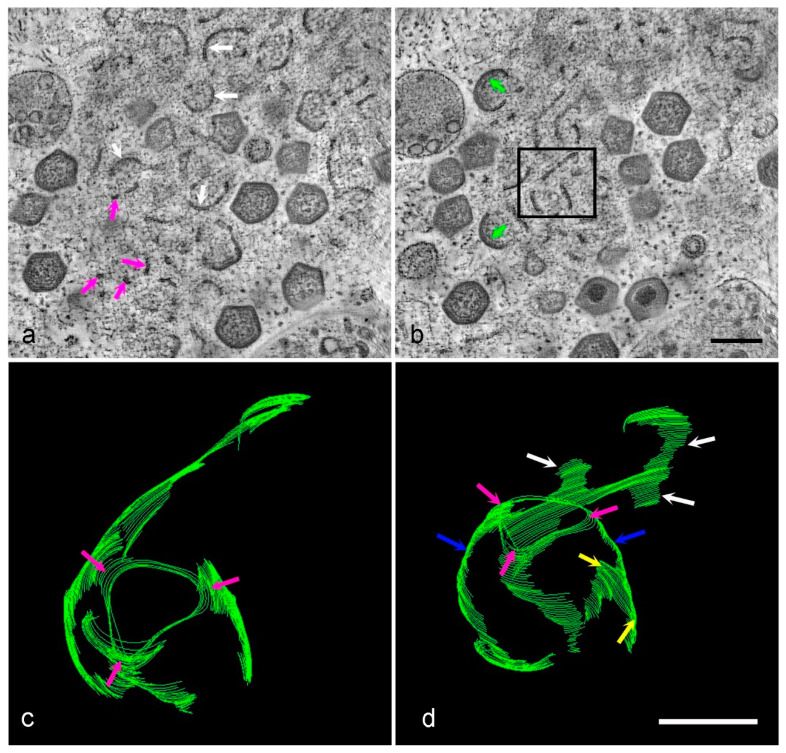
II. Tomographic reconstruction of HPF Vero cells at 18 hpi. (**a**,**b**) Two slices from the tomogram in Appendix A. All forms of membrane assembly intermediates are visible, membrane fragments (pink arrows), larger curved (uncoated) membranes (white arrows) and membranes coated in capsid and core proteins (green arrows). The area in the black box in (**b**) was modeled and viewed from two different angles (**c**,**d**). Intriguingly, this area appeared to contain a ‘skeleton’ virus with a triangular pole (**c**,**d**, pink arrows), vertical links down the body of the template (**d**, blue arrows) and horizontal spread of membrane at the center (**d**, yellow arrows). There appears to be a branched connection to one side (**d**, white arrows) which may link to other forming virions. Scale bars: b = 200 nm, d = 100 nm.

**Figure 7 viruses-13-00077-f007:**
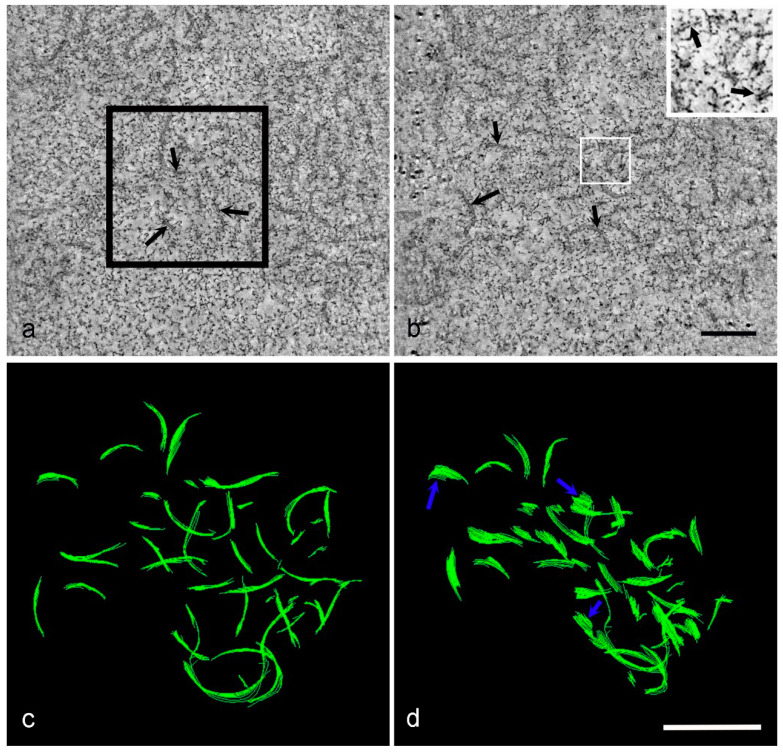
III. Tomographic reconstruction of HPF Vero cells at 18 hpi. Even though this infected cell was from the same sample as those in Figure 5 and Figure 6, it is clearly from a less mature factory. (**a**,**b**) Two slices from the tomogram in Appendix A. (**c**,**d**) The area in the black box in (**a**) was modeled and views from two angles are shown. Only early forms of the membrane assembly intermediates are visible in this factory, an enlarged image of the area in the white box in (**b**) shows the membranes more clearly after the brightness and contrast were adjusted. Membrane fragments (**a**,**b**, black arrows) and slightly larger curved membranes (**d**, blue arrows) appear to be in close association but un-linked. Intermediates with capsid and core protein deposits, immature and mature virions are not present in this region. Scale bars: b, d = 200 nm.

**Figure 8 viruses-13-00077-f008:**
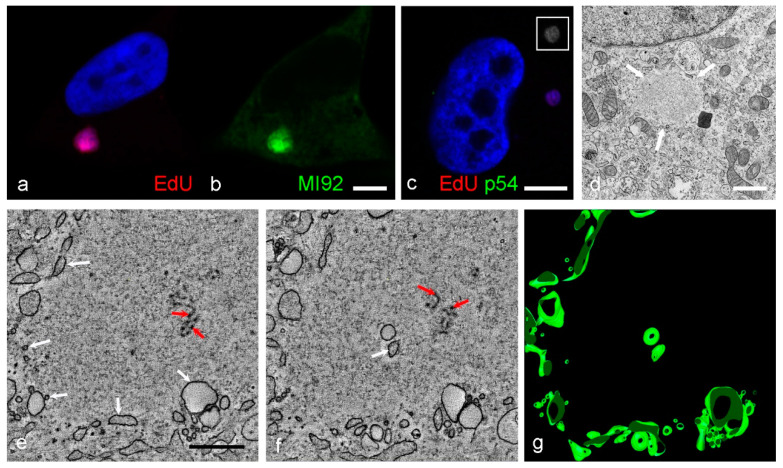
Early ASFV factories seen by confocal and electron microscopy. (**a**,**b**) Vero cells infected for 10 h had EdU incorporated into newly synthesized DNA, labelled with AlexaFluor 555 which localized to the viral factory. Labelling with an anti-ASFV serum (MI92) and AlexaFluor 488 highlighted the presence of viral protein, although this was shown not to be the intermediate protein p54 as anti-p54 antibody labelling (green) was negative at this early stage in infection (**c**). Area in the white box is a greyscale conversion (brightness enhanced) of the EdU labelling which is faint, but present. Cells infected for 12 h were prepared for TEM by chemical fixation. Cellular components have been cleared from an early factory (**d**, white arrows) although tomographic reconstruction (Appendix A) of a different factory from the same sample shows an accumulation of vesicles at the periphery (**e**, white arrows), two vesicles near the factory center (**f**, white arrow) and, separately, a small number of membrane fragments (**e**,**f**, red arrows). Vesicles are depicted in the model taken from this reconstruction (**g**). Scale bars: b, c = 5 µm, d = 1 µm, e = 500 nm.

## Data Availability

Data is contained within the article and Appendix A.

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
