# Peer review of "Unpicking the Secrets of African Swine Fever Viral Replication Sites"

_viruses, 2021, doi:10.3390/v13010077_

Round 1

Reviewer 1 Report

The manuscript requires minor revision.

The objective of this study was to analyse the viral factory with respect to the protein and membrane compartmentalization and organization. While the article provides interesting results, the manuscript requires revision before it can be acceptable for publication. There are some comments and suggestions listed below that should be considered.  

Line 117: „Cells were stained with DAPI or ToPro-3 (Life Technologies)...“ - Provide specific information on what staining method was used for what purpose and what the differences are.

Author Response

Response to Reviewer 1 comment:

The manuscript requires minor revision.

The objective of this study was to analyse the viral factory with respect to the protein and membrane compartmentalization and organization. While the article provides interesting results, the manuscript requires revision before it can be acceptable for publication. There are some comments and suggestions listed below that should be considered.  

We thank Reviewer 1 for their valuable comments, and the recommendation for publication with minor revision.

Line 117: „Cells were stained with DAPI or ToPro-3 (Life Technologies)...“ - Provide specific information on what staining method was used for what purpose and what the differences are.

This is an interesting point and one we should have explained initially.  For standard confocal microscopy we use DAPI to stain the DNA, not least because we have the corresponding filter set in our microscope stand to visualise the nuclei down the eyepieces while choosing cells to image.  However, we noticed very quickly that DAPI was not suitable for STED imaging because the 592 nm depletion laser excites it and causes a very high background.  Hence we choose to use ToPro-3 (ex 642 nm) to stain DNA when using STED superresolution.  Both stains are equally effective, in our hands, for imaging the cell nuclei and viral factories.  We have now explained this in the manuscript (lines 117 – 121).

Reviewer 2 Report

From the standpoint of a microscopist, I can confirm that this is a beautiful piece of work, since a number of challenging microscopic methods have been combined at a high methodological level. Superresolution STED (fluorescence light) microscopy was used to localize certain proteins in the viral factories. These data were combined with demanding electron microscopic preparation methods like high-pressure freezing and freeze substitution and tomographic three-dimensional electron microscopic imaging. This is a clever approach since STED allows for detection of certain proteins, whereas electron microscopy shows the whole ultrastructure of all cellular (and viral) components.

Therefore, I recommend accepting the manuscript for publication.

Some minor comments:

Line 15  „….replication sites knows as viral factories.” Maybe better “known as viral…”.

The insert in fig. 1a is barely visible. I think the fluorescence image should be brought to the same size as the DIC image. This also would make it easier to understand the message of the figure. Figs. 1g, h, and I are missing a magnification bar.

Line 320 “Fig. 5a and b are individual slices” I think the expression “virtual slices” instead of “individual slices” would be more accurate.

Author Response

Response to Reviewer 2 comments:

From the standpoint of a microscopist, I can confirm that this is a beautiful piece of work, since a number of challenging microscopic methods have been combined at a high methodological level. Superresolution STED (fluorescence light) microscopy was used to localize certain proteins in the viral factories. These data were combined with demanding electron microscopic preparation methods like high-pressure freezing and freeze substitution and tomographic three-dimensional electron microscopic imaging. This is a clever approach since STED allows for detection of certain proteins, whereas electron microscopy shows the whole ultrastructure of all cellular (and viral) components.

Therefore, I recommend accepting the manuscript for publication.

We thank Reviewer 2 for their kind words and recommendation for publication.

Some minor comments:

Line 15 „….replication sites knows as viral factories.” Maybe better “known as viral…”.

Thank you, this typographical error has been amended.

The insert in fig. 1a is barely visible. I think the fluorescence image should be brought to the same size as the DIC image. This also would make it easier to understand the message of the figure. Figs. 1g, h, and i are missing a magnification bar.

Yes, Reviewer 2 makes a very good point.  Figure 1 has been amended so that the DIC and DAPI images in panel a are the same size and proportion.  This has improved the figure.  A scale bar has been added to Figure 1i, representing all three zoomed areas (g, h and i) which are all the same size, as we have done for Figure 1d, e and f.

Line 320 “Fig. 5a and b are individual slices” I think the expression “virtual slices” instead of “individual slices” would be more accurate.

We agree with this comment and ‘individual’ has been changed to ‘virtual’.
